# The Use of Educational Games to Promote Learning in Geology: Conceptions of Middle and Secondary School Teachers

**Isabel Teixeira [1] and Clara Vasconcelos [1,2,*]**

1    Interdisciplinary Centre of Marine and Environmental Research (CIIMAR), University of Porto, Novo Edifício do Terminal de Cruzeiros do Porto de Leixões, Avenida General Norton de Matos, S/N, 4450-208 Matosinhos, Portugal; isabel.teixeira@fc.up.pt

2    Unit of Science Teaching & Department of Geosciences, Environment and Land Planning, Faculty of Sciences, University of Porto, Rua do Campo Alegre, S/N, 4169-007 Porto, Portugal

*    Correspondence: csvascon@fc.up.pt

**Abstract:** Studies highlight researchers' concerns about how science should be taught today. It is recognised that teachers have difficulty involving and motivating students to learn about different complex topics, such as geology. Schools must promote skills development to develop citizens who can be active and informed in society. One way of undertaking this is to use active methodologies such as educational games, in which students play an essential role in developing activities. Games encourage changes in conceptions regarding the relevance of this scientific topic that is often undervalued by students. Games have gained space in recent years in several disciplines, and it is essential that this tool is thought out and planned within a consistent pedagogical proposal. This educational resource is used to increase motivation for learning, as well as enhance and strengthen the effects of learning. An intervention plan can be framed within game-based teaching. Teachers have been underrepresented in the game-based learning literature, with more emphasis on games' effects. However, the pedagogical issue of games has been particularly relevant in recent decades. The current investigation used a survey given to geology teachers ($n = 112$) from public and private middle and secondary schools in Portugal. Its purpose was to assess teachers' perceptions regarding game-based teaching and its potential to promote active learning. Our sample ages ranged from 24 to 64 years (average of 48.9 years old); 81.3% were women and 18.8% were men. The analysis of the results seems to confirm that although they do not always use games to promote learning in geology, most teachers still recognise their potential to motivate, enhance, and reinforce the learning of geological content, with digital games being the preferred option. They emphasise the importance of teacher training in this area and the inclusion of game applications in school textbooks to approach different geology-related themes. Our results seem to indicate some lack of consistency in teachers' opinions on the impact of games on student learning.

**Keywords:** active learning; games; geology; teaching; survey

## 1. Introduction

Despite the crucial role of earth sciences in promoting citizens' scientific literacy and in understanding the environmental issues that have affected contemporary society, there is a gap between its significant relevance in society and the minimal emphasis placed on the topic in schools [1]. Thus, it is crucial to shift the focus of education to prioritise the development of competencies consistent with current societal demands and provide active environments to teach and learn. Moreover, there is a consensual understanding that traditional teaching methodologies do not entirely meet current demands, nor fulfil one of the most essential pillars of education: guaranteeing students' preparedness for the future [2]. Studies suggest the effectiveness of active methodologies compared to traditional ones [3]. In recent years, active learning methodologies have been widely adopted as pedagogical

processes in order to engage students in environments and activities to stimulate cognitive competencies and promote learning [4,5]. Active learning methodologies encourage students to become critical and interventional; these skills are needed to develop citizens who act as responsible members of society [6]. This teaching methodology should include strategies and tools that guarantee better learning effects and higher motivation [7]. Game-based learning aims to teach knowledge and competencies with games [8,9] and positively impacts problem solving and critical reasoning by providing joyful experiences, involvement, motivation, self-gratification, creativity, social interaction, and emotional stimulus [10,11]. Some authors also suggest that helping teachers understand the contribution of games to the learning process is essential to applying game-based teaching and achieving proposed goals [12,13]. As teachers are promoters of change in schools, they play a pivotal role in adopting and correctly implementing this teaching methodology [14]. Teachers are at the core of the teaching process and thus can promote the connection between game-based teaching and the curriculum, as well as evaluate the outcomes of this methodology [15,16]. However, they have been underrepresented in the game-based learning literature, with more emphasis on games' effects [17].

In the early 21st century, there was a notable surge in publications focused on games [18]. However, research indicates that European educators remain uncertain about the positive impact of games on developing competencies that should be acknowledged in students' performance across diverse subjects [19]. This underscores the existing scarcity of studies on game-based teaching, specifically in geology, and a lack of available games designed to enhance the learning experience in this scientific discipline. In our research, we discovered some geological games on Earth Learning Idea [20]. Among the games developed in Portugal, we identified Rock Cycle [21,22], Geological Race [23], Geogame [24], Shake Your Knowledge [25], Fossil Game [26], Minerals in your Daily Life [27], GEOtrivial Pursuit [28], and Minecraft [29]. Additionally, other games in the sample of this study are not widely known and, therefore, are rarely utilised. We refer to games such as those referenced in publications presented at the European Geoscience Union Congresses and in journal articles. They cover various geological topics, including geological hazards, volcanism, plate tectonics, earthquakes, geological time scales, and fossils. Most of them are digital games [30–34] and board games [35,36], while others involve cards [37] and role playing [38], and there are also some quizzes [39] and puzzles [40].

The present study aimed to investigate the opinions of geology teachers from public and private middle and secondary schools in Portugal on game-based teaching and its potential to promote active learning.

### 1.1. Games in Education

The boundaries between similar terms in this field (game-based teaching, serious games, didactic games, and others) are unclear. Ref. [41] defines a game as an activity that must have some specific characteristics, such as being fun, being limited in time and place, the outcome of the activity being unpredictable, being governed by rules, and being fictitious. Some authors [42] suggest other critical dimensions, such as sensory stimuli, challenge, mystery, and control. Games may be competitive or cooperative [43].

Different types of games with specific characteristics can be found in the literature. Some so-called entertainment games are not intended primarily for educational purposes [44]. These games are used as playful or gamified activities that provide a narrative context, challenge, or mystery in which students can engage [45]. The role of the teacher is important in this type of game, and more is required of them as there is no built-in pedagogical content, and the existing content may be incorrect or misleading [44].

Some authors [7] define an educational game as one designed and used for teaching and learning, in which we can combine elements of fun and educational concepts to increase students' motivation and engagement. Ref. [46] defines an educational game as a specific learning tool that requires learners to engage in competitive activities undertaken within rules. Another study [47] defines an educational game as a competitive activity governed

by rules which provide a connecting experience and understanding, allowing students a better grasp of the world. These games can be physical and digital.

Regarding the purpose of games, ref. [8] distinguishes educational games (merely a game) from didactic games (a game with a specific function in school learning). According to these authors, the first has an educational purpose applied in an informal way, regardless of place and time, with or without adult supervision. These games are designed for an educational purpose, and are intended to teach about certain subjects, increase the understanding of a historical or cultural event, or help students acquire competence while playing. The same study refers to didactic games specially designed for an instructive educational context that can be integrated and fully explored within this scope under the careful supervision and monitoring of teachers. The application of this didactic tool usually acts as a learning stimulus, prompting students to engage in lively discussion about the learning concepts after the game [7].

Some authors [48] consider so-called "serious" games that are specially created for teaching purposes and consider the discipline's subject matter. They, therefore, have a predetermined aim [49], being valuable tools not only for raising awareness about a topic or an issue, but also for promoting attitudinal or behavioural changes [50]. These games cover a wide range of genres, for example, simulations, strategy games, and adventure games [51]. Rather than entertainment, they seek to teach, increase engagement, and promote behavioural changes [52] and moments of reflection [53] on a given topic. They also allow students to observe, explore, recreate, manipulate variables, and receive immediate feedback on objects and events that would be too time-consuming, expensive, or dangerous to explore in a real setting [54]. They include digital and non-digital game formats [55].

### 1.2. Game-Based Teaching

In recent years, game-based teaching has been widely adopted as a teaching methodology [56]. In this methodology, students generally compete to achieve educational goals according to specific rules and principles, contributing to the development of their cognitive competencies and knowledge construction [57]. Game-based teaching is an innovative methodology likely to increase students' motivation, emotional involvement, and enjoyment [4]. As referenced, the findings of several studies suggest that this teaching methodology can effectively develop 21st-century competencies [7,58–60], such as critical thinking, creativity, collaboration, and communication [61].

For over twenty years, games have been used in education [7]; but, they are not sufficient for learning on their own. However, some elements can be activated within an instructional context that may enhance the learning process [42]. Some authors [62] provide several reasons why games are essential tools in teaching and learning. They consider that games (i) attract participation by students; (ii) can assist students in setting goals, ensuring goal rehearsal, and providing feedback, as well as reinforce learning and maintain records of behavioural change; (iii) can be used as research and/or measurement tools; (iv) allow the researcher to measure performance on a wide variety of tasks that can be easily changed, standardised and understood; (v) are fun and stimulating; (vi) allow students to experience novelty, curiosity and challenge, stimulating learning; (vii) help in the development of competencies; (viii) allow the examination of individual characteristics such as self-esteem, self-concept, goal setting, and individual differences; and (ix) can act as simulations. Games can also potentially involve the whole class in an active learning process [63]. Additionally, implementing games allows students to interact with educational materials, allowing a better understanding of learning outcomes [9] and possibilities for an interdisciplinary approach, as students can work on several competencies related to different disciplines. Thus, the application of games in teaching may perfectly fit the following objectives: enjoyable schooling—"science is fun"; applying useful information and competencies—"science is relevant"; and individual security—"science is a reliable and valuable investment" [64] (p. 404).

Notwithstanding the importance of game-based learning, the literature mentions some disadvantages, such as (i) the time involved making it difficult for the teacher to predict how much time students may need to complete all levels of a game and, thus, consider the game-related education task as completed; and (ii) the establishment of deadlines for students to finish a game, which can result in discouragement and low self-esteem for those who cannot complete the task in the given time [8].

Often, game-based teaching relies on technology such as computers, handheld devices, and online applications [4]. New technology and games are increasingly crucial in 21st-century education [65]. Today's schools deal with students born in a digital environment with different learning processes. Technological advancements have improved computer performance and the quality of graphics and sound resources, boosting the digital game industry's growth through various games adapted to various platforms. These digital tools are very popular among children and young adults [66,67]. Therefore, it is crucial to explore different ways of using this type of game to support learning in the classroom. The digital world has been primarily focused on building teaching–learning platforms and providing suitable learning materials [68]. Most games can take players into interactive virtual worlds, a reality that the teacher should consider. In a digitalised society with re-newed curricula, the meaningful integration of new tools and technology into teaching and learning depends on teachers' ability to (i) structure the learning environment in new ways, (ii) merge new technology with new pedagogy, and (iii) develop socially active classrooms encouraging cooperative interaction, collaborative learning, and group work [69]. Studies report a growing number of professionals and researchers who refer to digital games as a promising tool in teaching, as they enable student engagement and the development and mastery of essential competencies in the age of information and communication technology [70]. Digital games are characterised by several elements: entertainment, gameplay, rules, objectives, human–machine interaction, outcomes and feedback, adaptability, sense of triumph, conflict competitiveness and challenges, problem solving, social interaction, images, and narratives [71]. There are advantages over traditional learning materials, namely encouraging making and experimenting with different solutions for solving prob-lems and creating innovative teaching and learning scenarios in virtual and combined environments [72]. Thus, digital game-based learning connects the teaching process with new learning technologies, promoting cognitive changes and offering entertainment to students [8]. Additionally, it promotes motivation and willingness to learn and increases self-awareness [73]. Studies also refer to the added value of this resource in providing feed-back to students [74]. Some authors [19] underscored that teachers considered challenge, curiosity, pleasure, and cooperation the top four reasons for playing computer games to learn. However, some opponents to using these games in the classroom consider them merely a technological fashion emphasising superficial learning [18]. They believe that games are responsible for increasing violence, aggressiveness, inactivity, obesity, and de-creased pro-social behaviour [75]. However, such issues should not preclude using games in school learning [8].

## 2. Materials and Methods

This research aimed to investigate the perspectives of middle and secondary school geology teachers on the effectiveness of game-based teaching in promoting meaningful learning. A survey was conducted using a questionnaire comprising 20 questions. The participants included geology teachers from middle school to the end of secondary school, specifically those teaching natural sciences (grades K5 to K9), biology and geology (grades K10 and K11), and geology (grades K12) in both Portuguese public and private schools. The survey was distributed through the Google Forms platform and typically required approximately 10 min for completion, although there was no specified time limit for respondents.

## 2.1. Validation and Reliability of the Questionnaire

The questionnaire underwent content validation to ensure its accuracy and relevance. The questions aligned with the literature review's findings and were subjected to content validation by two researchers in science education. Additionally, the survey's reliability was meticulously addressed. A team of researchers aged 37 to 65 possessed considerable experience, with an average of 16 years of service extending almost to the end of their careers (with retirement at 67). The researchers consisted of two females and two males. Among them, two were professors in higher education specialising in geoscience education, while the remaining four were educators in biology and geology at middle school and secondary levels. This iterative process of reliability involved reading, answering, and improving the questions until a consensus was reached among all team members. To further enhance reliability, the team members completed the questionnaire twice, with a 10-day interval between the first and second administrations. The final version of the questionnaire encompassed five sociodemographic questions, thirteen multiple-choice questions, and two open-ended questions.

## 2.2. Ethical Considerations

In this study, ethical procedures inherent in the field of social sciences were employed. Specific legislation in the study's country ensured participants' data protection. Participants were fully informed about the study's objectives. In the introduction to the online questionnaires, participants were presented with the following statement: "I hereby declare that I agree to provide data for this survey, and I am aware that my data will be used for analytical purposes by the study and further publishment. I agree that my data will be retained at least until the completion of the survey analyses and destroyed when it is no longer necessary for the study's aims." This statement, along with the corresponding response in the questionnaire, was considered as informed consent.

## 2.3. Sample

We employed a convenience sample of middle and secondary school teachers (*n* = 112) from Portuguese public and private schools. Further details regarding the sample can be found in Table 1.

**Table 1.** Sociodemographic profile of the sample.

| Gender | Female (*n* = 91) | Male (*n* = 21) |
| --- | --- | --- |
| Age | M 49.1<br>Min–Max 24–64 | M 48.3<br>Min–Max 27–63 |
| Employment relationship | C (*n* = 23; 25.3%)<br>E (*n* = 68; 74.7%) | C (*n* = 5; 23.8%)<br>E (*n* = 16; 76.2%) |
| Years of service | 0–10 (*n* = 14; 15.4%)<br>11–20 (*n* = 19; 20.9%)<br>21–30 (*n* = 34; 37.4%)<br>≥31 (*n* = 24; 26.4%) | 0–10 (*n* = 2; 9.5%)<br>11–20 (*n* = 2; 9.5%)<br>21–30 (*n* = 11; 52.4%)<br>≥31 (*n* = 6; 28.6%) |
| Type of school | P (*n* = 82; 90.1%)<br>Pr (*n* = 7; 7.7%)<br>P and Pr (*n* = 2; 2.2%) | P (*n* = 19; 90.5%)<br>Pr (*n* = 1; 4.8%)<br>P and Pr (*n* = 1; 4.8%) |
| Subjects and levels currently taught | 2CN (*n* = 8; 8.8%)<br>3CN (*n* = 40; 44.0%)<br>BG (*n* = 20; 22.0%)<br>2CN3CN (*n* = 4; 4.4%)<br>3CNBG (*n* = 19; 20.9%) | 2CN (*n* = 2; 9.5%)<br>3CN (*n* = 7; 33.3%)<br>BG (*n* = 8; 38.1%)<br>2CN3CN (*n* = 1; 4.8%)<br>3CNBG (*n* = 3; 14.3%) |

Note: M, mean; Min, minimum; Max, maximum; C, contractor; E, effective; P, public school; Pr, private school; 2CN, natural sciences (K5 to K6); 3CN, natural sciences (K7 to K9); BG, biology and geology (secondary); 2CN3CN, natural sciences (K5 to K9); 3CNBG, natural sciences (K7 to K9) and biology and geology (secondary).

*2.4. Procedure*

Data were systematically gathered through a comprehensive 20-question online questionnaire (see Appendix A) strategically disseminated across the social networks of professional groups specialising in natural sciences, biology, and geology. This network included a projected total of 4300 teachers. The acquisition of 112 responses took 2 months. The 112 completed questionnaires were obtained from respondents whose ages ranged from 24 to 64, with an average age of 48.9. The participant demographic consisted of 81.3% women and 18.8% men. According to an annual national report, Portugal has the oldest teaching staff in the European Union and is not producing enough new teachers [76]. The average age of teachers in Portugal (excluding those in higher education) is 50.2 years. The Retirement and Pensioners List reveals that 262 kindergarten teachers and 3259 middle and secondary school teachers retired in 2023. Compared to the previous year, there is a notable increase of 46.6%, as in 2022, 2401 educators from the public education system of the Portuguese Ministry of Education retired. This scenario may have contributed to the disinterest of teachers in participating in research studies.

*2.5. Data Analysis Techniques*

The quantitative data collected in this study were analysed utilising IBM® SPSS® software, specifically version 28, tailored for the statistical analysis of questionnaires. The statistical analysis included descriptive measures such as frequencies, means, Cramer's V, and Eta coefficient. A chi-square test was also performed. Notably, the choice of these statistical tests was guided by the study's objectives and the inherent nature of the variables under scrutiny.

**3. Results**

The results will be presented by analysing each question from the survey. Questions Q1 to Q6 referred to sociodemographic aspects of the sample already presented in Table 1.

Regarding whether they utilised games to enhance geological learning (Q7), 92 teachers affirmed this, constituting 82.1% of the respondents.

An examination of the association between Question 7 and various factors, including age, gender, employment relationship, years of service, school type (public or private), subjects taught, and current teaching levels, was conducted. The measures of association, Eta, and the symmetric measure Cramer's V are detailed in Table 2. There was minimal association between the variable (Q7) and the other variables (Q1–Q5), with a weak association between Q7 and Q6.

**Table 2.** Eta coefficient and symmetric measure Cramer's V concerning Q7 and Q2–Q6.

| Questions | Association and Symmetrical Measures | *p*-Value |
|:---:|:---:|:---:|
| Q7/Q1 | Eta | 0.035 |
| Q7/Q2 | Cramer's V | 0.045 |
| Q7/Q3 | Cramer's V | 0.054 |
| Q7/Q4 | Eta | 0.031 |
| Q7/Q5 | Cramer's V | 0.076 |
| Q7/Q6 | Cramer's V | 0.216 |

Regarding participants who did not use games to promote geological learning (*n* = 20; 17.9%), when queried about the associated constraints with utilising games in geological teaching (Q8), the majority cited a lack of awareness of the existence of this tool (*n* = 13; 65.0%). Additionally, some respondents mentioned class characteristics (*n* = 3; 15.0%), while a few participants reported that games have an infantilising effect on the teaching/learning processes (*n* = 2; 10.0%), impeding the progression of the essential discipline/learning program (*n* = 2; 10.0%).

It is important to note that questions 9 through 13 were directed only to teachers who responded affirmatively to Question 7.

Regarding the benefits of using games (Q9), most teachers ($n$ = 69; 75.0%) highlighted their motivating, enhancing, and strengthening impact on learning. Seventeen teachers (18.5%) believed that games motivated students to learn, contributing to increased participation and engagement. A small percentage of teachers selected all the listed benefits in the questionnaire options ($n$ = 6; 6.5%).

As for the frequency of using games (Q10), 48 participants (52.2%) reported "sometimes," 21 mentioned "often" ($n$ = 21; 22.8%), 19 teachers indicated "rarely" ($n$ = 19; 20.7%), and only a few selected "always" ($n$ = 4; 4.3%).

Regarding typology, digital games emerged as the most preferred ($n$ = 50; 54.3%), with 36 participants (39.1%) indicating using both digital and physical games. Conversely, only six participants (6.5%) opted for physical games (Q11).

In Question 12, where teachers were asked to name games they frequently used, 60 teachers (65.2%) mentioned computer and/or application games. Other options were cited by a limited number of respondents, such as 10 (10.9%) employing board games and games in applications and/or computers, 8 (8.7%) selecting card games and games in applications and/or computers, and 6 (6.5%) mentioning memory games and games in applications and/or computers. Two teachers (2.2%) chose all the options in the questionnaire, and only one teacher (1.1%) selected a diverse combination of board and card games, games in applications and/or computers, challenge games, word games, and questions and answers.

For assessing the impact of games on students' learning (Q13), teachers favoured grids ($n$ = 51; 55.4%), while some utilized reports ($n$ = 14; 15.2%), and others employed tests ($n$ = 10; 10.9%). However, 17 teachers (18.5%) reported not using any instrument to assess the impact. No differences were found between years of service and whether teachers used an instrument to measure the impact of the game on student learning ($\chi^2$ = 0.171; $p$ = 0.679).

Regarding Question 14, most teachers ($n$ = 108; 96.4%) considered using games as a teaching and learning strategy. Three teachers neither agreed nor disagreed (2.7%), and only one disagreed (0.9%).

Regarding the question about the playful nature of games (Q15), a notable number of respondents expressed disagreement ($n$ = 40; 35.7%), while the agreement option received a comparable number of responses ($n$ = 38; 33.9%). Additionally, 24 teachers neither agreed nor disagreed (21.4%), while 7 (6.3%) totally agreed and 3 (2.7%) totally disagreed.

Responses varied in response to the inquiry about the most suitable schooling levels for learning-oriented teaching through games (Q16). Thirty-three teachers (29.5%) favoured middle education; an equal number indicated middle and secondary education. Fifteen teachers (13.4%) suggested K7–K9, twelve (10.7%) considered K7–K9 and secondary, eleven (9.8%) mentioned K5–K6, and four (3.6%) selected secondary education. One teacher (0.9%) opted for K5–K6 and secondary, and three teachers (2.7%) answered "none".

While the majority of teachers ($n$ = 98; 87.5%) agreed that games could contribute to the development of students' critical thinking and raise awareness of the importance of geology in everyday life (Q17), four participants disagreed (3.6%), and ten teachers neither agreed nor disagreed (8.9%).

Question (Q18) aimed to explore which geological topic(s) is (are) best suited for teaching through games. Significantly, 39.3% ($n$ = 44) of teachers considered all topics suitable. The answer "Consequences of the Earth's internal dynamics" was chosen by 29 teachers (25.9%), while 31 teachers (27.7%) selected other themes. It should be noted that five teachers (4.4%) answered that they did not know, and three (2.7%) answers were excluded for not addressing the question.

Question 19 addressed whether teachers considered it relevant to have training on teaching geared towards learning through games. The majority answered affirmatively ($n$ = 96; 85.7%), while some respondents neither agreed nor disagreed ($n$ = 10; 8.9%), and only six teachers disagreed with this statement (5.4%). Teachers also advocated for school textbooks to include game proposals addressing various geological topics (Q20). The results

revealed that most participants agreed (*n* = 99; 88.4%), with very few disagreeing (*n* = 5; 4.5%). Some teachers neither agreed nor disagreed (*n* = 8; 7.1%).

## 4. Discussion

This study aimed to examine the opinions of teachers from public and private middle and secondary schools in Portugal on game-based teaching in geology and its potential to promote learning. The participants reported positive attitudes toward using games and said that this pedagogical approach contributes to learning. Furthermore, the results showed that most teachers recognise the capabilities of games, even if they do not use them in their teaching practice.

The literature shows teachers' increasing use of games to address different subjects [9]. Therefore, not surprisingly, most teachers in the present study mentioned using games to promote geological learning (Q7).

Question 8 was only directed at participants who did not use games. The constraints that teachers associated with using games in teaching geology were not coincident with the disadvantages [8] suggested by the literature. For example, studies indicate that European teachers are not yet sure about the positive effects of games on the development of some competencies and students' performance in various subjects [19], thus representing barriers to the adoption of games in the classroom [77].

In the present study, most participants referred to the main benefits of using games (Q9), such as their motivating effect and enhancing and reinforcing learning. The literature [7,57,78] stresses this potential, although the benefits of games are considerably wider [79].

The answers to Question 10 suggest that teachers do not regularly use games. The majority of the teachers said that they only use games occasionally. These results are aligned with the speciality literature [80] that also states that teachers still do not use games regularly, especially in geology.

The results are not surprising (Q11), as some studies suggest that digital games have the potential to be a learning tool [70,80]. Perhaps most participants preferred this type of game for this reason. Nevertheless, research [79] has highlighted the limited application of digital games in education because of teachers' distrust of the usefulness of this tool.

Again, it was also not surprising to verify that games in apps and/or computers were the most chosen (Q12), as mentioned in the previous question and consistent with the literature. In addition, regarding physical games, refs. [81–83] highlight more geosciences board game options than others. This fact may explain teachers' preference for board games over other physical games. In addition, the literature refers to studies that highlight the importance of this type of game in enhancing cognitive function [84].

Although grids were mainly used by teachers in assessing a game's impact on student learning (Q13), the number of participants who reported not using any assessment instrument is worth mentioning. The low percentage of respondents who reported not using instruments to assess the impact of games on students' learning is relevant in considering the potential of this resource and its further assessment. Studies [17] highlight the importance of games in assessment. It should be noted that the literature reports [18] the benefits of assessment, namely that tracking motivational, emotional, and metacognitive characteristics will help to better understand specific behaviours and final outcomes.

The analysis of the results of Question 14 suggests that the teachers participating in this study recognise using games as a teaching and learning strategy [4]. Notwithstanding, these results are not aligned with the fact that a significant percentage of teachers do not carry out any evaluation instrument (previous answer) for games played in the classroom.

The number of teachers who believe that games are primarily used for recreational purposes was found to be identical to the number who disagreed with this statement (Q15). These results indicate some discrepancy in teachers' opinions regarding the playful nature of this resource. However, a slightly higher number of teachers did not consider the main function of games to be their playfulness. Interestingly, the literature highlights the playful

features of games, among other characteristics [41]. However, other studies argue that, rather than entertaining, this resource should teach, increase engagement, and promote "players'" behavioural changes [52].

From the respondents' perspective (Q16), school levels K7–K9 and K5–K6 are best suited to learning-oriented teaching through games, although many teachers mentioned secondary education as well. These results align with the literature, which states that game-based learning can be applied to enhance students' learning processes in various age groups [7].

Question 17 explored whether games can contribute to developing students' critical thinking and raising awareness of the importance of geology in everyday life. Most teachers agreed with this statement. This answer is consistent with the literature, which states that game-based learning enhances the learning experience while maintaining a balance between the game content and its application to everyday life [85]. Studies also suggest that games can effectively develop 21st-century competencies such as critical thinking [7,58–60]. It should be emphasised that earth sciences must prepare students to become participative citizens and leaders of change [86], and games can be a tool that should be considered by teachers to achieve this.

Curiously, many teachers believed that all geological topics are suitable for learning-oriented teaching through games (Q18). Furthermore, studies mention that this recourse has gained space in different subjects [9]. These results strongly suggest that this tool may also be appropriate for geological topics.

Most teachers emphasised the importance of having teacher training focused on learning through games (Q19). A study with teachers from Italy, Turkey, and the UK found that training teachers in game-based learning was not considered a priority [87]. More recently, practical training in this area was part of a research study with Montenegrin teachers, who began to use their competencies to create games in their classrooms [88]. Another recent study found that teacher training had a positive impact on teachers' ability to identify both the benefits and barriers of game-based learning [89]. This type of action could help overcome the barriers mentioned by teachers in Question 8, where they reported being unaware of the existence of games for the content they teach. In addition, it could help teachers realise the importance of assessing a game's impact on students' learning.

Most of this study's participants agreed that school textbooks should include proposals for games that address various geology-related topics (Q20). These findings support the notion that textbook editors can suggest this resource in their educational projects. However, notwithstanding the importance of this topic to the teaching/learning process, only a few proposals were found for geoscience games in Portuguese textbooks.

## 5. Conclusions

The current study's primary objective was to investigate a convenient sample of geology teachers' perspectives regarding game-based teaching and its potential to facilitate active learning in middle and secondary schools. The survey results indicate that most Portuguese geology teachers view games as an effective teaching and learning strategy across various educational levels and geological topics, with digital games emerging as the preferred option. The analysis of the results suggests that while teachers may not consistently employ games to promote geological learning, they generally still recognise their potential to motivate, enhance, and reinforce the understanding of geological content. As a result, game-based teaching emerges as a recommended integral component of geology's teaching and learning processes. The identified significance of games as tools for promoting geological learning, along with the insights gleaned from the administered questionnaire, underscores the importance of teacher training in this domain. Furthermore, it advocates for including game applications in school textbooks, offering a diverse approach to various geological themes. Based on these study results, professional development based on game-based methodologies should be conducted with pre-service and in-service teachers. These studies will help to verify if game-based methodology allows teachers to design, implement,

and assess students after engaging with educational games. Learning will also be an asset, allowing us to conclude the benefits and potential of this methodology in teaching and learning processes. Teacher training will be crucial in fostering an understanding among educators of the significance of employing assessment instruments tailored to game-based learning. Teachers often lean towards traditional grids, especially when they are easy to use. However, it is essential that assessment instruments are meticulously developed, validated, and possess reliability to be viable for use in research. Nevertheless, the survey responses highlight game-based teaching methodology as a practical approach for fostering active learning. Further research is warranted to substantiate the inclusion of this methodology in geology curricula for 21st-century educational practices.

**Author Contributions:** Conceptualization, I.T. and C.V.; formal analysis, I.T. and C.V.; investigation, I.T. and C.V.; methodology, I.T. and C.V.; software, SPSS treatment, I.T. and C.V.; resources, I.T.; supervision, C.V.; validation, C.V.; writing—original draft, I.T.; writing—review and editing, I.T. and C.V. All authors have read and agreed to the published version of the manuscript.

**Funding:** This research was supported by national funds through FCT—Foundation for Science and Technology—within the scope of UIDB/04423/2020 and UIDP/04423/2020. It was also supported by the National Agency for Financing Science, Research and Technology (FCT) through a Doctoral Scholarship (ref. 2022.11255.BD).

**Institutional Review Board Statement:** The study did not require ethical approval.

**Informed Consent Statement:** All teachers who answered the questionnaire gave informed consent.

**Data Availability Statement:** Datasets are with the first author.

**Conflicts of Interest:** The authors declare no conflicts of interest.

## Appendix A. Questionnaire on the Use of Games in the Classroom

This questionnaire is intended to assess the geology teacher's opinion on teaching geared to learning through games. The time it takes to fill in the questionnaire is short and only addressed to secondary Natural Sciences teachers (K5 to K9) and Biology and Geology, and Geology teachers (K10 to K12). This questionnaire is voluntary and anonymous.

We thank you for your availability and the sincerity of your answers. Your participation validates the consent for using responses in developing this study.

* Required:

"I hereby declare that I agree to provide data for this survey, and I am aware that my data will be used for analytical purposes by the study and further publishment. I agree that my personal data will be retained at least until the completion of the survey analyses and destroyed when it is no longer necessary for the study's aims." *

**Q1—Age** *

**Q2—Gender** *
Female/Male/Other

**Q3—Professional Status** *
Contractor/effective

**Q4—Length of service (in years)** *

**Q5—School where you teach** *
Public/private school/Both

**Q6—Subject(s) you currently teach (Select one or more options)** *
Natural Sciences (K5 to K6)
Natural Sciences (K7 to K9)
Biology and Geology (Secondary)
Geology (Secondary)

**Q7—Do you use, or have you used games to promote the learning of geology? \***
Yes/No

**Q8—If you answered negatively, what constraints do you associate with using games in teaching geology? (Select one or more options) \***
It makes it challenging to comply with the essential discipline/learning program.
It infantilises the teaching and learning processes.
I do not recognise the potential in its application.
I am unaware of the existence of games for the content I teach.
Other

**Q9—If you moved to this question, you answered affirmatively to the previous question. What benefits do you attribute to the application of games in teaching geology? (Select one or more options) \***
It motivates the student to learn.
It reinforces peer interaction.
It enhances student participation and involvement in classes.
It develops student autonomy.
It improves school success measured in tests.
Other

**Q10—How often do you use the game? \***
Rarely/Sometimes/Often/Ever

**Q11—What kind of game do you prefer when teaching geology? \***
Physical/Digital/Both

**Q12—Of the following examples, which games do you use most often? (Select one or more options) \***
Cards game/Board game/Memory game/Game in applications and/or computer/Other

**Q13—What instrument do you use to measure the game's impact on student learning (grid, test, report, etc., no instrument)? \***

**Q14—The use of the game is a teaching and learning strategy. \***
I totally disagree/I disagree/I neither agree nor disagree/I agree/I totally agree

**Q15—First and foremost, the game must always have a playful function. \***
I totally disagree/I disagree/I neither agree nor disagree/I agree/I totally agree

**Q16—Which level(s) of schooling do you consider best suited to teaching geared to learning through games? (Select one or more options) \***
K5 to K6/K7 to K9/Secondary/None

**Q17—Games can contribute to the development of students' critical thinking, making them aware of the importance of geology in everyday life. \***
I totally disagree/I disagree/I neither agree nor disagree/I agree/I totally agree

**Q18—Which geology topic(s) is (are) best suited to teaching geared to learning through games? \***

**Q19—I consider it relevant to train teachers on teaching geared to learning through games. \***
I totally disagree/I disagree/I neither agree nor disagree/I agree/I totally agree

**Q20—It is important that school textbooks include game applications in approaching different geology-related topics. \***
I totally disagree/I disagree/I neither agree nor disagree/I agree/I totally agree

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
