# Peer review of "The Use of Educational Games to Promote Learning in Geology: Conceptions of Middle and Secondary School Teachers"

_geosciences, doi:10.3390/geosciences14010016_

Round 1

Reviewer 1 Report

Comments and Suggestions for Authors

The authors of the article "Educational Games to Promote Geosciences Learning: Conceptions of Middle and Secondary Teachers" (I. Teixeira and C. Vasconcelos) introduce the subject of the application of games in teaching. The article presents a literature review and an illustrative test case (geosciences). The drafting is straightforward, the language is clear, the methodology is appropriate, and the findings are well presented.

The drawback of the article is the imbalance between the detailed literature review and a very elementary example. The essence of the observational findings is that teachers need better training in using games, even if the teachers take a favourable view on using games for educational purposes. The article should be restructured, bringing this insight to the forefront.

Further comments:

1.       The authors should clarify what subjects they include in the notion of geosciences. Considering the text and the questionnaire [Q18, Q20], they seem to think mainly about geology. Geosciences is a much broader concept – see, for example, the disciplines covered by the Europen Geosciences Union.

2.       Some information about the ‘validators’ of the questionnaire (line 228), e.g., age, gender, and experience, would be helpful.

3.       The methodological section should mention the likely size of the audience (line 259ff)  to whom the questionnaire was presented. The (low ?) number of responses should be assessed. Possibly, also considering the observation: ‘reduced number of geosciences games’ (line 360).

a.       However, see:

                                                             i.      https://blogs.egu.eu/geolog/tag/geoscience-games/

                                                            ii.      https://blogs.egu.eu/geolog/tag/gaming/

4.       The study focuses on teachers using games (lines 276, 286). This feature should be stressed upfront. The skewness of the study to this sub-population is relevant given the essential finding (= more training needed).

As a detail (lines 91-96): The order of statements should be altered.

Summary; the strength of the article is indicating: ‘games are important for teaching; however geoscience teachers need more training to make good use of this tool’.

The question remains whether the study's finding was to be expected and whether its publication should wait until a more substantial observational base has been gathered. 

Author Response

Dear Reviewer

We want to express our gratitude for the time dedicated to reading and reviewing our manuscript. We have taken into consideration the suggestions we gave and made the choices we made. All comments addressed to reviewers are distinctly highlighted in red throughout the text.

Please take a look at the attachment.

Season's greetings

Reviewer 2 Report

Comments and Suggestions for Authors

This is a properly written paper but I am giving it a low review because I think it does not address the main problems with geoscience education as stated in the introduction.  Citizens of the world need to know more about the Earth and their environment than is commonly taught in schools so that they can make intelligent decisions about the world around them.  That said, games as a teaching tool may be useful and I would like to see more information about some of the available games used to teach geoscience.   My concern is that the teachers are using games because the teachers are poorly prepared to teach geoscience and sending students to a computer to be entertained while learning material the teachers may not be able to make interesting otherwise is not the solution.  The nice thing about digital learning is that the teacher does not have to prepare the material or grade it so it relieves a poorly trained teacher from some of the teaching responsibilities and improves the student outcomes.

In the U.S. most natural science teachers have degrees in either education or Biology.  Few have ever taken a course in geology.  Most of those teachers emphasize learning definitions in geology because that is what they know how to teach.  The environmental approach usually focuses on Biology even though the earth should be a critical part of that lesson.  This survey would have been more useful if teacher preparation had been addressed and correlated with the responses.  Understanding the motivation of the teachers in responding to the questions asked is essential—that is, are they ducking their teaching responsibilities because they are poorly prepared or are they truly good teachers who want to motivate their students. I do not think that having more digital games to do the teaching of geosciences is a better solution than training teachers better to teach geoscience.  I am also concerned that the survey sample was too small and the reported support for using gaming to teach geoscience was not as well supported in many cases as suggested in the discussion.  The stated “critical thinking and motivational improvements” described by the authors were only based on the opinions of the teachers not on any measurement of those factors.

I think this study has potential if it is redesigned to address my concerns and it does indicate that many teachers would us digital games in their teaching if they had more information and training.  I would certainly like to see a better designed study on this topic because gaming is a valuable teaching tool in my opinion.

Author Response

(The authors gave the same response as above.)

Reviewer 3 Report

Comments and Suggestions for Authors

Dear Authors, I have read and carefully evaluated your manuscript and I found it interesting, addressing a relevant topic and well inside the aims and scopes of the journal. However, I recommend some revisions before publication. Please, take into consideration my comments below.

GENERAL COMMENTS

- English is generally good but some sentences are problematic. For example, the first and the third sentences of the abstract are rather obscure to me, they need to be rephrased. 

- The text is long and it contains several repetitions. Moreover (especially in the introduction) paragraphs are sometimes composed by short sentences (followed by a reference) not well logically connected to each other. I think you should go through the text and shorten it consistently: less is more, and in scientific papers concepts usually are not repeated twice. 

- I think it is mandatory to make some examples about the games used by the teachers. It is surprisingly that they are not clearly mentioned. Whoever reads the paper will be curious to know which games are available and which have been used. 

- Similar to the previous point, I was surprised that the introduction has a detailed part about games in education, one about game-based teaching and nothing specific about games in geosciences, which is the topic of the paper. I think it is mandatory that you make at least some examples connected to some key game gategories such as boardgames (https://doi.org/10.5194/nhess-16-135-2016), role playing games (https://doi.org/10.1088/2515-7620/ac6f47), quiz (https://doi.org/10.3390/geosciences13110322), video games (https://doi.org/10.5194/gc-5-325-2022 , https://hdl.handle.net/2268/264934)

SPECIFIC COMMENTS

L26: this seems a contradiction: "they do not always use games [...] they resort to this tool". Please, clarify.

Keywords have some issues. Try to avoid keywords composed by many words. 

L91: which terms?

Section 2.3 It would be nice to know how geographically distributed are the participants to the questionnaire. Do they all come to big cities? 

Section 2.4 I think you should provide the complete list of the questions. If you don't want to put it in the text (for reasons of space) you can provide an annex as a supplementary material. I think it is better to read the questions now, instead of discovering them when you discuss the answers. 

References: please check the formatting. Sometimes italics is on the journal, sometimes on the title.   

Comments on the Quality of English Language

English is fine. Maybe some minor edits are needed: they could be addressed during the typesetting stage. 

Author Response

(The authors gave the same response as above.)

Round 2

Reviewer 1 Report

Comments and Suggestions for Authors

-none-

Author Response

Dear Reviewer

The authors appreciate the review, especially during this festive time. Thanks for the acceptance. Happy holidays and a joyful 2024!

Reviewer 2 Report

Comments and Suggestions for Authors

this is a significant improvement over the first version.  In particular the regular emphasis on the objective of measuring teachers opinions about using games rather than the value of games in promoting student learning was good.  I am not excited by this paper but it is now sufficiently useful that I would exam whether I should encourage the use of games more strongly in my professional development courses.

Author Response

(The authors gave the same response as above.)

Reviewer 3 Report

Comments and Suggestions for Authors

Dear Authors, thank you for addressing my comments. 

I liked the manuscript since the first version, now it has been improved even more. I think it is ready for publication.

I have just one single concern.

In the revised version you substituted the term "geosciences" with "geology".

Probably it was the other reviewer that asked this change! I have to say that I do not agree with this change.

First of all, geosciences is a broader term that may save you from misinterpretations (see e.g. second point).

Second, and most importantly, when you write "geology learning", "geology teachers", "geology students" it sounds strange to me: I doubt in Portugal there is a high school subject called "geology". In all European country I am aware of, "Earth Sciences" or "geosciences" are taught, not "geology". If this is the case also in Portugal, I strongly recommend turning again to "geosciences". 

Author Response

Dear Reviewer

The authors appreciate the review, especially during this festive time.

The change from 'geoscience' to 'geology' was suggested by one of the reviewers, but it's a suggestion that we consider to be a better option. In Portugal, there is indeed a discipline called 'Biology-Geology' as a mandatory subject in grades K10-K11, and the discipline of 'Geology' as an optional subject in K12. Therefore, we have retained the term 'geology' accordingly. In the attached PDF document, we have included the national curriculum where you can verify the mentioned subjects.

Thanks for the acceptance. Happy holidays and a joyful 2024!
